# Auditory-GAN: deep learning framework for improved auditory spatial attention detection

Tasleem Kausar[1], Yun Lu[2], Muhammad Awais Asghar[1], Adeeba Kausar[3], Siqi Cai[4], Saeed Ahmed[5] and Ahmad Almogren[6]

[1] Electrical Engineering, Mirpur Institute of Technology, Mirpur University of Science and Technology, (MUST), Mirpur, AJK, Pakistan
[2] Computer Science, School of Computer Science and Engineering, Huizhou University, Huizhou, Guangdong, China
[3] Computer Science, Department of Information Engineering Technology, Superior University, Lahore, Punjab, Pakistan
[4] Electrical and Computer Engineering, Department of Electrical and Computer Engineering, National University of Singapore, Singapore
[5] Electrical Engineering, Department of Electrical Engineering, Mirpur University of Science and Technology (MUST), Mirpur, Pakistan
[6] Department of Computer Science, College of Computer and Information Sciences, King Saud University, Riyadh, Saudi Arabia

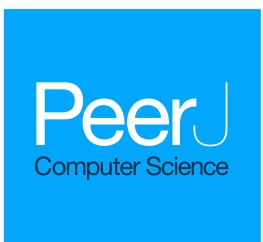

Corresponding author
Yun Lu, luyun_hit@163.com

## ABSTRACT

Recent advances in auditory attention detection from multichannel electroencephalography (EEG) signals encounter the challenges of the scarcity of available online EEG data and the detection of auditory attention with low latency. To this end, we propose a complete deep auditory generative adversarial network auxiliary, named auditory-GAN, designed to handle these challenges while generating EEG data and executing auditory spatial detection. The proposed auditory-GAN system consists of a spectro-spatial feature extraction (SSF) module and an auditory generative adversarial network auxiliary (AD-GAN) classifier. The SSF module extracts the spatial feature maps by learning the topographic specificity of alpha power from EEG signals. The designed AD-GAN network addresses the need for extensive training data by synthesizing augmented versions of original EEG data. We validated the proposed method on the widely used KUL dataset. The model assesses the quality of generated EEG images and the accuracy of auditory spatial attention detection. Results show that the proposed auditory-GAN can produce convincing EEG data and achieves a significant *i.e.*, 98.5% spatial attention detection accuracy for a 10-s decision window of 64-channel EEG data. Comparative analysis reveals that the proposed neural approach outperforms existing state-of-the-art models across EEG data ranging from 64 to 32 channels. The Auditory-GAN model is available at https://github.com/tasleem-hello/Auditory-GAN-/tree/Auditory-GAN.

## INTRODUCTION

Auditory attention is a cognitive process in which the brain selectively focuses on specific auditory stimuli while filtering out irrelevant or distracting sounds in a multi-speaker situation, commonly termed "cocktail party scenario" (*Mesgarani & Chang, 2012*). This ability to selectively focus on specific sounds is critical for daily communication, but individuals with hearing loss face challenges in such environments (*Das, Francart & Bertrand, 2019*). Several auditory assistive devices developed with noise suppression algorithms often struggle to accurately select the targeted speaker in cocktail party scenarios, hindering speech enhancement.

Recent neuroscience research shows that auditory attention may be realized directly from brain signals such as electrocorticography (EEG) data (*O'Sullivan et al., 2015*; *Mirkovic et al., 2015*; *Das, Francart & Bertrand, 2019*; *Das, Bertrand & Francart, 2018*; *Zhao et al., 2018*) termed auditory attention detection (AAD). The aforementioned findings from studies inspire researchers to create new hearing aids known as neuro-steered hearing aids, that improve the target speaker by explicitly decoding attention-related features from brain signals. Several AAD approaches for brain-computer interface (BCI) have been published. Multiple studies have investigated the responses and speech stimuli for AAD. For instance, *O'Sullivan et al. (2015)* employed a stimulus reconstruction method considering the decoding of the speech envelope corresponding to the attended speaker. Brain signals are utilized to approximate the speech envelope perceived by the subject and speaker bearing an elevated correlation coefficient, considered the attended speaker. In literature, different variants of the stimulus reconstruction algorithms have been projected to increase the AAD efficiency (*Das, Francart & Bertrand, 2019*; *Miran et al., 2018*; *de Cheveigné et al., 2018*; *Aroudi & Doclo, 2020*; *de Taillez, Kollmeier & Meyer, 2020*).

However, stimulus reconstruction-based AAD decoders still experience several imitations. To achieve vital AAD, the temporal resolution of stimulus reconstruction techniques is typically around ten to tens of seconds, which is considered practically inapplicable for real-time BCI operations (*Stegman et al., 2020*; *Abiri et al., 2020*; *Robinson, Chester & Kg, 2021*). Mainly, uncertainties in correlations among the rebuilt and the actual speech envelopes happen when computed over a small window length that contains little speech information (*Robinson, Chester & Kg, 2021*; *Geirnaert, Francart & Bertrand, 2020*). However, humans can shift their attention from one target speaker to another with a temporal resolution of approximately 1 s. Therefore, it is challenging for the stimulus reconstruction based AAD decoders to perform the AAD over such a high temporal resolution. Likewise, the stimulus reconstruction methods require clean speech signals to perform reliable auditory attention detection, which limits their use in real-world applications. For example, in the process of voice acquisition through robots, a system is anticipated to execute tasks in complicated acoustic surroundings of multiple speakers. In several existing studies, speech separation algorithms (*Van Eyndhoven, Francart & Bertrand, 2017*; *Das et al., 2020*; *Ceolini et al., 2020*) have been proposed to extract speech envelopes for AAD systems. The demixing process adds to the computation burden and

increases the computational cost, leading to decreased efficiency of the AAD models, and limiting their success in real-life applications (*Geirnaert, Francart & Bertrand, 2021a*).

Motivated by the above findings, it is anticipated that the spatial location of auditory attention is neurally encoded (*Kerlin, Shahin & Miller, 2010*; *Frey et al., 2014*; *Wöstmann et al., 2016*; *Bednar & Lalor, 2020*; *Deng, Choi & Shinn-Cunningham, 2020*; *Geirnaert, Francart & Bertrand, 2021b*). The hypothesis can be set that the locus of auditory attention can be cracked from neural/brain activities. This paradigm can perform auditory attention detection without speech stimulus envelopes, making the AAD exceedingly capable for neuro-steered hearing aids to achieve high efficiency exclusive of clean speech stimuli as the source. Recently, a convolutional neural network (CNN)-based approach has been developed by *Vandecappelle et al. (2021)*, which decoded the location of auditory attention directly from EEG signals. This approach achieved a high detection accuracy of 80.8% for the 1-s decision window, but the multivariate temporal information had not yet been effectively exploited in the spatial domain. In another study, authors designed a spatio-spectral feature (SSF) representation-based AAD method that outperformed the up-to-date models in AAD tasks. *Vandecappelle et al. (2021)* used the alpha power (8–13 Hz) of EEG signals to identify the spatial attention (left and right). The topographical specificity of alpha power indicated the direction of auditory attention to speech (*Kerlin, Shahin & Miller, 2010*; *Frey et al., 2014*; *Bednar & Lalor, 2020*; *Deng, Choi & Shinn-Cunningham, 2020*). *Cai et al. (2022)* used the SSF representation and employed the spiking neuron model to design a neural-stimulated EEG-based AAD system. This model performed auditory attention detection from EEG without a clean speech envelope.

To further improve spatial auditory detection accuracy, another SSF study (*Jiang, Chen & Jin, 2022*) takes into account multiple frequency bands to extract the spectro-spatial feature (SSF) for AAD instead of relying on a single frequency band.

Several studies have highlighted the effectiveness of GCNs for EEG-based emotion recognition (*Ceolini et al., 2020*; *Geirnaert, Francart & Bertrand, 2021a*) sleep stage classification (*Kerlin, Shahin & Miller, 2010*), motor imagery classification (*Frey et al., 2014*) and epileptic seizures detection from EEG signals (*Saminu et al., 2022*). *Wöstmann et al. (2016)* employed the GCN model in EEG-based AAD task to obtain state-of-the-art detection performance.

In recent studies, the effectiveness of graph convolutional networks (GCNs) has been studied for spatial auditory attention detection using EEG signals. *Cai, Schultz & Li (2024)* designed an EEG-Graph Net model based on brain topology for the EEG-enabled AAD task. They achieved state-of-the-art results by employing a neural attention mechanism. However, this study mainly relied on predefined graph structures derived from prior knowledge, such as 2D or 3D spatial positions of EEG electrodes. More recently, the AGSLnet framework has been designed (*Zeng, Cai & Xie, 2024*). This model leverages latent relationships between EEG channels to improve AAD performance.

However, high-accuracy detection of auditory attention within small time intervals remains challenging. It is also well-established that the detection performance of deep learning-based methods is affected by the quality and quantity of training data (*Mokayed et al., 2022*). The advent of deep learning models, particularly convolutional neural

networks (CNNs) (*Liu et al., 2022a*, *2022b*; *Li et al., 2021*) requires vast training data volumes to enhance performance while preventing overfitting. The deep models, especially convolution neural networks, consist of a sequence of hidden layers and millions of free parameters that need to be well-trained with abundant data. However, a large public EEG training dataset is not available online. The lack of online available brain data for AAD tasks led us to explore ways of expanding EEG datasets.

Data augmentation is another technique used to artificially generate the dataset based on real-time existing data employing various transformations. Data augmentation is widely used in machine learning and plays a vital role in improving performance (*Liu et al., 2023a*, *2023b*; *Zhang et al., 2023a*), reducing overfitting, and reinforcing the generalization capabilities of deep models (*Chlap et al., 2021*; *Garcea et al., 2023*; *Li, Hou & Che, 2022*). The GANs and their variants have shown potential in the image (*Zhang et al., 2023b*; *Han, Liu & Chen, 2023*; *Kausar et al., 2023*), artificial audio (*Donahue, McAuley & Puckette, 2019*), and electroencephalographic (EEG) brain signals generation (*Xu et al., 2022*). Initially, the GAN model (*Goodfellow et al., 2020*) was developed to generate artificial image datasets. It has the prominent capability to synthesize augmented versions of original images to enrich available training datasets. Nevertheless, the original vanilla GAN network (*Goodfellow et al., 2020*) encounters challenges with training stability, and detailed regularization is required to achieve desirable performance. In this regard, several research approaches have been projected to achieve the GAN network training stability and improve its generation efficiency.

Several variants of traditional GAN networks have been investigated. For instance, Wasserstein GAN (WGAN) (*Gulrajani et al., 2017*) employed a new loss function by using the Wasserstein distance formula that contributes to improving the model stability and quality of generated images. They proposed an improved training strategy to ensure stable training of various GAN architectures. *Radford, Metz & Chintala (2016)* designed deep convolutional generative adversarial networks (DCGANs) for representations of objects. *Chang, Chen & Chung (2018)* established a semi-supervised learning-based GAN to generate class labels in a discriminator network and develop the quality of generated images. *Odena, Olah & Shlens (2017)* proposed a new variant of GAN called auxiliary classifier GAN (ACGAN) which used label information and synthesized high-quality images to achieve robust performance in classification tasks.

However, GAN applications in AAD for data augmentation are still unexplored. In this article, we proposed an auditory-GAN system consisting of an SSF module and an auditory generative adversarial network auxiliary (AD-GAN) classifier. Integrating AD-GAN and SSF representation is an innovative work in this context aiming to revolutionize the research in auditory attention detection. The proposed auditory-GAN framework represents a groundbreaking approach toward addressing the challenges of data scarcity and enhancing the classification of auditory spatial attention in EEG signals. The experimental analysis reveals that the proposed auditory-GAN system generated convincing EEG data and obtained a significant performance. Our designed auditory-GAN system model is a promising tool for real-time auditory spatial attention detection, with potential applications in brain-computer interfaces and neurofeedback systems.

The key contributions of this paper are as follows:

- We introduce a novel auditory-GAN system comprising two modules: the SSF module and an end-to-end AD-GAN classifier. The SSF module extracts spatial feature maps by capturing the topographic specificity of alpha power from EEG signals, while the AD-GAN network mitigates the need for extensive training data by synthesizing augmented versions of the original EEG data.
- We implemented and validated our model, comparing its performance with other state-of-the-art auditory attention decoders. The results confirm the effectiveness of our proposed approach.

## METHODOLOGY

As illustrated in Fig. 1, the proposed auditory attention detection system, *i.e.*, AD-GAN, consists of two essential modules: an SSF extraction module and an AD-GAN network. The SSF module extracts the spatial features by learning the topographical specificity of alpha power from EEG signals (*Frey et al., 2014*; *Cai et al., 2021*). The designed AD-GAN network generates the augmented version of EEG data and performs auditory spatial attention detection (left/right), as depicted in Fig. 2. In the first phase, we designed an SSF extraction module to perform the feature extraction, as well as to combine spectral and spatial features from the EEG alpha strip.

In the SSF extraction process, the fast Fourier transform (FFT) is applied on a continuous time series related to each electrode, and the power spectrum of the EEG signal is computed. Subsequently, the measurement value of each electrode is found by computing the average squared absolute value in the frequency band. To exploit the potential of the spatial features of EEG signals, measurements of various decision windows are converted into a sequence of 2-D images. The Azimuth Equidistant Projection technique (*Snyder, 1987*) is applied to project all the EEG electrodes from the 3-D space onto a 2-D plane. To ensure that all points are precisely spread out from the center, all points are projected onto a plane tangent to the earth, and all latitude and longitude lines are split into identical parts. Likewise, to represent the spatial distribution of EEG signals, a $28 \times 28$ mesh using the Clough-Tocher interpolant method is employed to evaluate each grid (*Amidror, 2002*).

This approach enables the generation of topographical activity maps from EEG signals, obliterating the requirement of manual feature crafting. These maps depict the alpha frequency band in specific time windows (*Deng, Choi & Shinn-Cunningham, 2020*), referred to as SSF maps hereafter. The SSF maps, obtained from successive time windows, capture the temporal evolution of brain actions. The SSF maps, presented as 2-D images, act as input to the succeeding AD-GAN network.

In the second phase, the SSF maps, representing "real EEG images," are inserted into the proposed AD-GAN network to generate extensive EEG training data. A standard GAN network comprises two modules, the generator (G) and the discriminator (D), which are trained competitively (*Goodfellow et al., 2020*). The GAN network leverages the adversarial networks to enhance the quality of generated images. The proposed scheme adopted and

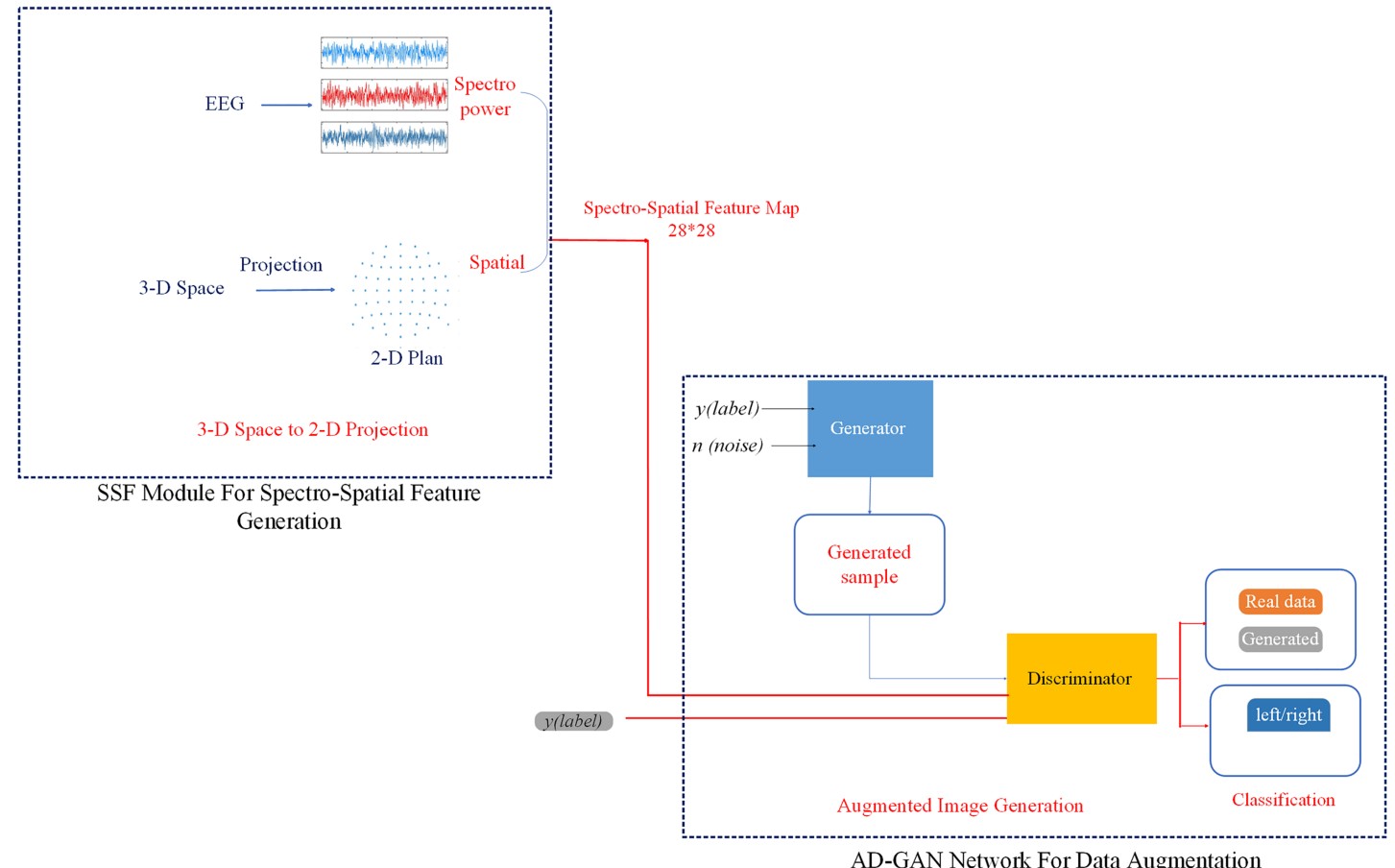

**Figure 1 Workflow diagram of the proposed auditory-GAN system consisting of an SSF module and an auditory generative adversarial network auxiliary (AD-GAN) classifier.** The auditory-GAN system works in three stages: Spectro-spatial feature map formation, a. The 2D images shown in the picture are spectro-spatial feature maps generated from EEG signals using the SSF module. They were not directly downloaded from any online dataset. EEG signal dataset is available at *Das, Francart & Bertrand (2019)*. 

modified the ACGAN network (*Odena, Olah & Shlens, 2017*), a variant of standard GAN (*Goodfellow et al., 2020*) that mixes class conditional architecture and supplementary network for categorization. The ACGAN (auxiliary classifier generative adversarial network) employs additional class labels to train the network while improving the excellence of generated image samples. Furthermore, the discriminator in ACGAN has an auxiliary part to generate specific class labels. This class conditional generation process in the ACGAN discriminator recognizes the generated images and differentiates the various classes. In contrast to traditional GANs, ACGAN can generate the best-quality images and deliver label data concurrently.

The proposed AD-GAN network is based on the latest ACGAN network (*Odena, Olah & Shlens, 2017*). Mathematically, both random noise $n$ and image labels $y$ are fed to the generator module in the AD-GAN model to produce augmented EEG image samples $X_g = G_{(n,y)}$, and discriminator output probability values on input EEG images and class

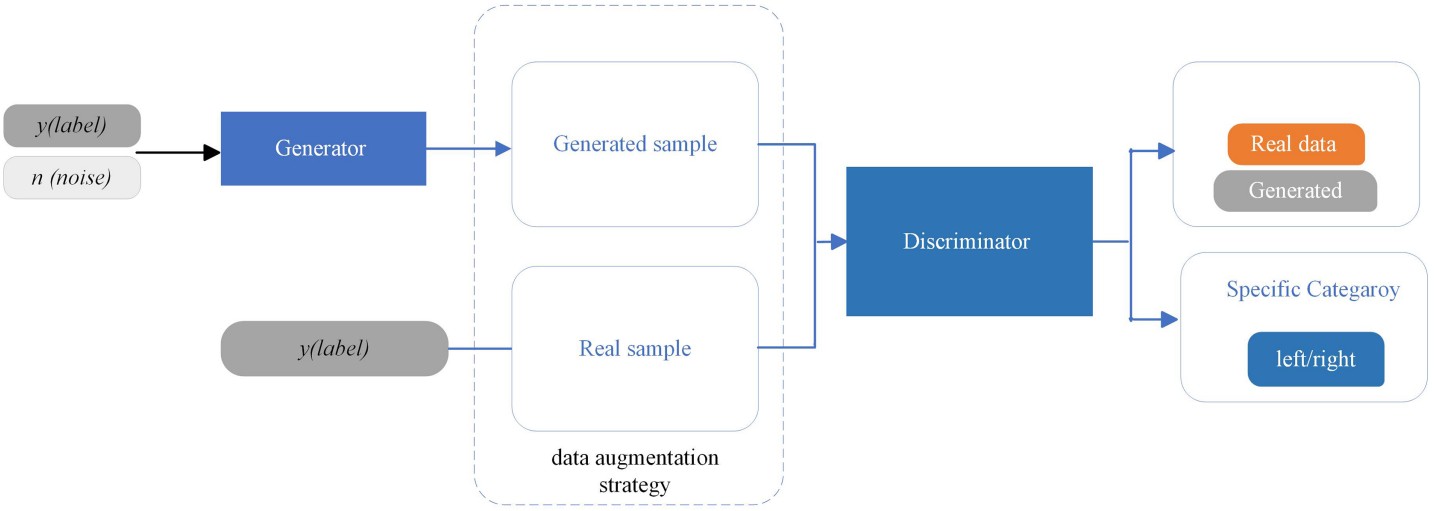

**Figure 2** The designed AD-GAN network based on an auxiliary classifier generative adversarial network.

labels. The given objective function in Eq. (1) comprises two log-likelihoods related to the correct image trace.

$$L_{source} = E_{x:P_{data}}[\log P(s = real|x_r] + E_n \sim P(n)[\log P(s = generated|x_g)] \tag{1}$$

$$L_{class} = E_{x:P_{data}}[\log P(Class = c|x_r)] + E_n \sim P(n)[\log P(Class = c|x_g)] \tag{2}$$

where $P_{data}$ denotes the actual distribution of the data, $P(n)$ past distribution on noise vector $n$, $E_{x \sim P_{data}}$ is the expectation of $x$ from real data distribution $P_{data}$, and $E_n \sim P(n)$ denotes expectation of $n$ sampled from noise.

The discriminator is trained to maximize log likelihood $L_{class} + L_{source}$, while for the generator, the training goal is to maximize $L_{class} - L_{source}$ loss. The designed architecture of AD-GAN shows superiority in generating high-quality image samples. This augmentation shows that the AD-GAN is particularly well-suited for EEG image augmentation and auditory attenuation detection. The workflow diagram of the proposed AD-GAN system is given in Fig. 2.

The detailed architecture of our designed AD-GAN network, consisting of the generator and the discriminator, is given in Table 1. The generator produces the images from latent space with distinctive labels, aiming to fool the discriminator. Real EEG images, *i.e.*, SSF maps, from the KUL AAD dataset (*Ji et al., 2022*) and generated EEG images by the designed AD-GAN network, are shown in Table 1. Before feeding the data to the discriminator, generated image data and real training data are combined. For each input image, the discriminator generates two distinct labels: one to determine whether the image is actual (output 1) or generated (output 0) and another to identify the specific category it belongs to (left or right auditory spatial attention). After generating the artificial versions of EEG images, the dataset is augmented using generated and real images. In the next

**Table 1 Real and generated samples.** (A) Real EEG images (SSF maps) from the KUL AAD dataset; (B) Generated EEG images by designed AD-GAN model. A simple visual analysis reveals an approximate similarity between the generated and real images. The 2D images shown in the picture in (A) are spectro-spatial feature maps generated from EEG signals using the SSF module. They were not directly downloaded from any online dataset. In (B) the shown pictures are the SSF maps generated using AD-GAN model. EEG signal dataset is available at *Das, Francart & Bertrand (2019)*.

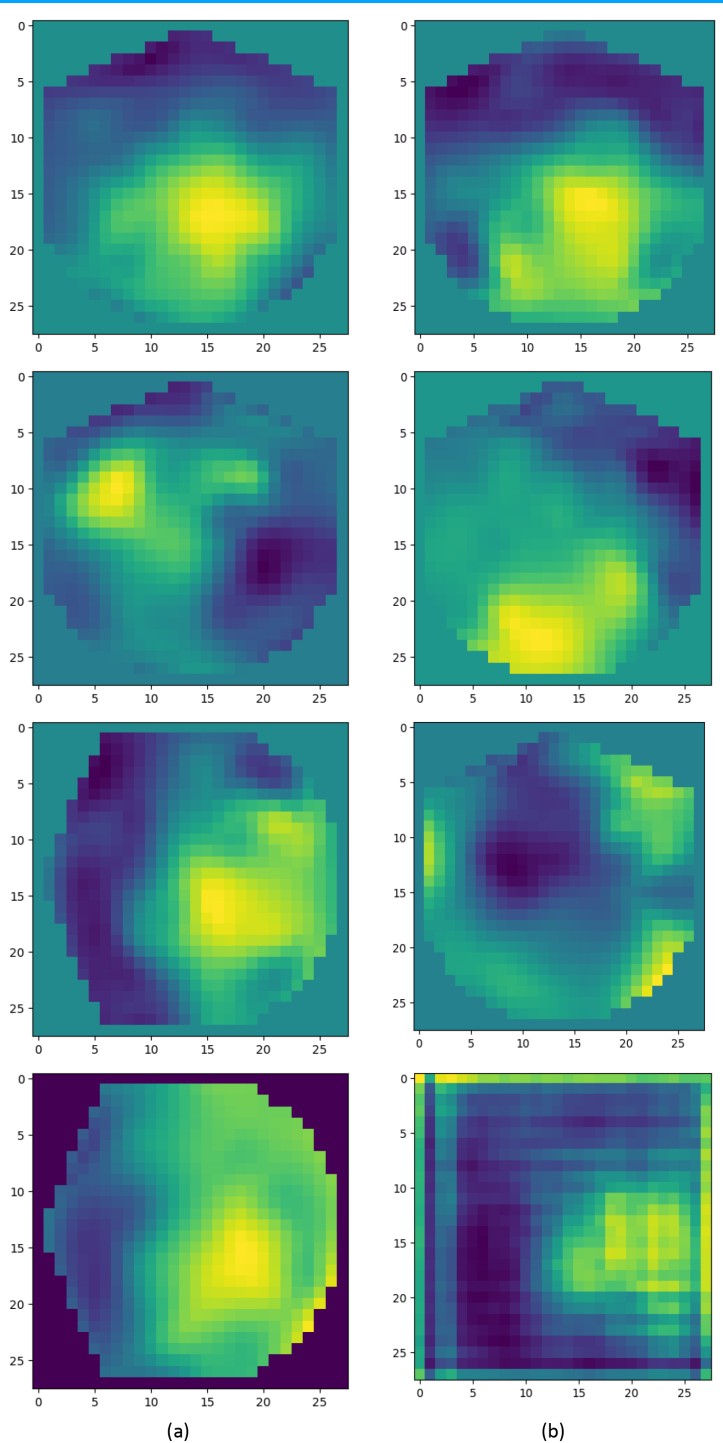

(a)                                      (b)

**Table 2** The design hierarchy of the proposed AD-GAN network consisting of (A) generator and (B) discriminator.

| Layer | Output shape | Parameters |
| --- | --- | --- |
| (a) | | |
| FC | (12544) | 1266944 |
| BN | (12544) | 50176 |
| Activation | (12544) | 0 |
| Reshape | (7, 7, 256) | 0 |
| Dropout | (7, 7, 256) | 0 |
| Conv2D Transpose | (7, 7, 128) | 819328 |
| BN | (7, 7, 128) | 512 |
| Activation | (7, 7, 128) | 0 |
| Up Sampling2D | (14, 14, 128) | 0 |
| Conv2D Transpose | (14, 14, 64) | 73792 |
| BN | (14, 14, 64) | 256 |
| Activation | (14, 14, 64) | 0 |
| UpSampling2D | (28, 28, 64) | 0 |
| Conv2D Transpose | (28, 28, 32) | 18464 |
| BN | (28, 28, 32) | 128 |
| Activation | (28, 28, 32) | 0 |
| Conv2D Transpose | (28, 28, 1) | 289 |
| Activation | (28, 28, 1) | 0 |
| (b) | | |
| Conv2D | (14, 14, 64) | 640 |
| Leaky ReLU | (14, 14, 64) | 0 |
| Dropout | (14, 14, 64) | 0 |
| Conv2D | (7, 7, 128) | 73856 |
| LeakyReLU | (7, 7, 128) | 0 |
| Dropout | (7, 7, 128) | 0 |
| Conv2D | (4, 4, 256) | 295168 |
| LeakyReLU | (4, 4, 256) | 0 |
| Dropout | (4, 4, 256) | 0 |
| Conv2D | (4, 4, 512) | 1180160 |
| LeakyReLU | (4, 4, 512) | 0 |
| Dropout | (4, 4, 512) | 0 |
| Flatten | (8192) | 0 |
| FC (Activation: sigmoid) | (1) | 8193 |

phase, the condition of the generated data is gaged by applying both the real and generated data to the real problem, *i.e.*, auditory spatial attention detection. The statistical characteristics are inspected, and experimental analysis is executed to ascertain the efficacy of the produced data.

**Table 3 Hyper-parameter settings for model training on the KUL dataset of 64 channels using a decision window of 10 s.**

| Experiments | Learning rate (Generator) | Learning rate (Discriminator) | Batch size | Epochs | Accuracy (%) |
|---|---|---|---|---|---|
| 1 | 0.0001 | 0.0002 | 256 | 90 | 95.5 |
| 2 | 0.0003 | 0.0002 | 64 | 130 | 96.2 |
| **3** | **0.0002** | **0.0001** | **8** | **200** | **98.5** |

The architecture of the proposed AD-GAN network consisting of the generator and the discriminator is given in Table 2. The 2D-dimensional convolution operation is employed to build the AD-GAN network. Convolutional operations can extract local features from images, multi-layered architecture of convolutions can learn the hierarchical representations. Unlike unsupervised learning algorithms, class-based conditional neural networks use the auxiliary data of the category labels that assist in model convergence. In the proposed AD-GAN network, Batch normalization is utilized to mitigate the effect of gradient vanishing during model training. The generator is designed to generate images using a latent vector 'n' drawn from a uniform distribution $(-1, 1)$, represented as n $(-1, 1)$. The images produced by the generator are augmented versions of the original data. The augmented images along with original images are fed to the discriminator. In the discriminator, the sigmoid function is employed to predict sampled images, and SoftMax function to predict specific labels. For the proposed AD-GAN network, model parameters are updated iteratively based on the loss function given in Eqs. (1) and (2). To analyze the effect of hyperparameters, several experiments were performed with different values. These hyperparameter settings led to optimal performance. The hyperparameter values and model accuracies are listed in Table 3. After multiple rounds of experiments, the best hyperparameter values were selected. In the first, second, and third experiments, the model was trained for 90, 130, and 200 epochs, respectively. The best performance was achieved when the training was performed for 200 epochs using a batch size of eight and a learning rate of 0.0001 for the discriminator and 0.0002 for the generator with the ADAM optimizer. The training process is carried out in the following steps. In the first step, the generator generates image samples from random noise using specific labels. In the second step, generated images and original images are mixed and input to the discriminator. In the third step, the discriminator is trained using the augmented image data and labels while parameters are updated iteratively. The discriminator architecture starts its training process after the training is complete. Finally, the parameters of the discriminator are frozen, and it remains untrainable. During this process, only parameters in the generator are updated and trained to generate more accurate EEG image samples. Once the combined structure has been trained (after finishing epoch iteration) the training process starts again from the first step, and multiple training iterations are repeated to balance the discriminator and generator. After enough iterations, both the generator and discriminator losses reach Nash Equilibrium enabling the generator to generate more realistic images when provided with specific labels.

# DATASET AND EXPERIMENTAL SETTINGS

## EEG dataset and data preparation

For the analysis of the performance of the designed system, the auditory attention detection analysis is performed using the KUL AAD dataset (*Ji et al., 2022*). The 64-channel KUL ADD dataset is acquired from 16 normal-hearing individuals consisting of eight males and eight females. The EEG data was collected in a soundproof and electromagnetic-shielded room where the subject paid attention to one of two competing speakers. The 64-channel EEG data is acquired using BioSemi Active Two device. The sampling frequency is offset at 8,192 Hz. The electrode positioning that adheres to the international 10–20 system is set. A low-pass filter of 4 kHz cut-off frequency is applied to auditory signals and then delivered at a 60 dB volume level using a pair of in-ear earphones (Etymotic ER3). The audio stimuli were displayed dichotically (one speaker per ear) or simulated using head-related transfer function (HRTF) filtering with two speakers coming from 90 degrees to the left and the right of the subject, respectively. During the experiments, the arrangement of presentation for both conditions was randomized to different subjects.

A total of 20 trials are conducted for each subject, the first eight trials are of 6 min duration while 12 trials are of 2 min duration. In order to make a fair comparison with the existing state-of-the-art methods, we performed experiments on the data collected in the first eight trials. Therefore, 12.8 h of EEG data for all 16 subjects ($8 \times 6$ min = 48 min of EEG data per subject) is collected and used in our experiments.

**Data preparation:** Each trial is filtered using high pass with a 0.5 Hz cut-off frequency and then down sampled to 70 Hz. In the data preprocessing stage, the data segments are generated using a sliding time window method of five durations $T = 0.5, 1, 2, 5,$ and $10$ s with an overlap of 50%. Therefore, with a decision window of 0.1 s, 5,760 decision windows are generated in the test set for each subject, accumulating 92,160 decision windows. In addition, zero mean and unit variance is retained by, normalizing the dataset channels for each trial. To prepare for the SSF extraction from EEG signals, the SSF module is implemented utilizing MATLAB R2014a. The AD-GAN model is executed employing the widely used TensorFlow (*Abadi et al., 2016*) and Keras (*Chollet, 2018*) libraries. A PC Intel®Xeon®CPU E5-1620 v3 PC with 3.54.00 GHz is used. With a memory of 12 GB NVIDIA Tesla M40 GPU is employed.

**Performance evaluation:** We divide the data into training data (80%), and testing data (20%). For the detection of auditory spatial attention, the proposed AD-GAN network not only uses the real training data to train the model but also uses the generated samples for training. In the testing phase, the testing dataset is employed to validate the AD-GAN network to observe the quality of generated samples. The testing data is not training procedure. To optimize the model learning, the ADAM optimizer bearing 32 batch size is used. The following metrics are computed to mainly measure the performance in the detection of the proposed auditory-GAN model.

**Table 4  Statistical analysis with conventional GAN (*Goodfellow et al., 2020*) and proposed auditory-GAN networks.**

| Model | ED | PCC | K-LD |
|---|---|---|---|
| SSF+GAN | 0.1384 | 0.8256 | 0.1576 |
| **SSF+AD-GAN (auditory-GAN)** | 0.0077 | 0.8987 | 0.1465 |

$$Precision = \frac{TP}{TP + FP} \tag{3}$$

$$Recall = \frac{TP}{TP + FN} \tag{4}$$

$$F\ score = 2 * \frac{Recall * Precision}{Recall + Precision} \tag{5}$$

$$Total\ accuracy = \frac{\sum TP}{Total\ images} \tag{6}$$

where TP, FP, and FN represent true positives, false positives, and false negatives, respectively.

## RESULTS AND DISCUSSION

**Generated data evaluation and application in auditory attention detection:** After training the auditory-GAN model, it generated artificial versions of the original training images. The generated image and real samples are shown in Table 1. From a simple visual analysis, we can approximately find the similarity between generated and real images. A statistical analysis is performed to determine the excellence of generated images. The Euclidean Distance (ED) metric (*Dokmanic et al., 2015*) is measured, which shows the gap between generated EEG images and the original EEG images, smaller value indicates more similarity between generated and original images, and high value indicates less similarity. To check the distribution consistency between generated and original images the Kullback–Leibler Divergence (K–LD) metric (*Ji et al., 2022*) is computed. The higher K–LD value shows more divergence between the two distributions which is an indication of worse performance. In contrast, the percentage of consonants correct (PCC) metric (*Shriberg et al., 1997*) is computed to check the correlation between the generated samples and the original image. A high value of PCC over 0.8 represents a strong similarity between generated and original images. The values given in Table 4 are an indication of the better performance of our proposed scheme in comparison to other existing schemes which represents that the generated images are much closer to the real ones.

To inspect the efficacy of the AD-GAN model in enhancing AAD (when training data is limited), we have performed numerous experiments with diverse sets of data. In these experiments, the effect of data quantity on model learning is confirmed corrodingly. In various scenarios, different sizes of original and generated datasets et and details of the setting of each scenario have been shown in Table 5. The detection performance of each model in the mentioned circumstances after adequate training has been given in Table 5.

**Table 5 Effect of data augmentation on auditory-GAN detection accuracy using a decision window of 10 s.**

| Real training images | 12,896 | 12,896 | 12,896 | 12,896 | 12,896 |
|---|---|---|---|---|---|
| Generated images | 0 | 35,792 | 48,688 | 61,584 | 84,480 |
| Accuracy (%) | 75.6 | 79.5 | 81.9 | 90.8 | 98.5 |

**Table 6 Detection accuracy (%) under various proportions of available real data using a decision window of 10 s.**

| Further generated samples | Existing real data | | |
|---|---|---|---|
| | 25% | 50% | 100% |
| 0 | 65.8 | 68.9 | 75.6 |
| 35,792 | 67.9 | 71.5 | 79.5 |
| 48,688 | 69.6 | 73.5 | 81.9 |
| 61,584 | 70.3 | 76.2 | 90.8 |
| 84,480 | 74.2 | 87.9 | 98.5 |

**Table 7 The accuracy achieved using different traditional augmentation techniques.**

| Experiments | Techniques | Accuracy (%) |
|---|---|---|
| 1 | Original set | 75.6 |
| 2 | Rotated images | 79.3 |
| 4 | Scaled images | 74.2 |
| 5 | Original + rotated + scaled | 85.2 |

The model trained on the larger scale data shows better performance. From the results given in Table 5, it is evident that the model trained on the mixed data (real + generated) achieves highly accurate predictions in contrast to the model trained only on generated data. For enough training iterations, the model trained with generated data achieves superior performance.

Several scenarios explored by employing several ratios of the existing training data are explained in the following. Classification analysis is given in Table 6 under distinctive settings of training data. With various proportions of training data, we used different numbers of generated samples. As well as comparative analysis is performed without data augmentation. From the results given in Table 6, it is evident that when the training data is sufficient, AD-GAN based data augmentation strategy does not significantly influence the performance, there is small improvement in classification accuracy. The performance improvement is obvious, when available training data proportion drops to 20%, which confirms the efficacy of the proposed method. Our GAN-based strategy for data augmentation starts a concern in producing raw EEG samples in auditory attention detection.

**Table 8 Detection accuracy (%) across five different decision window sizes on the 64 channel KUL dataset.**

| Model | Auditory stimulus | Decision window | | | | |
|---|---|---|---|---|---|---|
| | | 0.1 s | 1 s | 2 s | 5 s | 10 s |
| Linear decoder (*O'Sullivan et al., 2015*) | With | – | 58.1 | 61.3 | 67.5 | 75.8 |
| CNN (*Vandecappelle et al., 2021*) | Without | 65.9 | 80.8 | 82.1 | 83.6 | 85.6 |
| SSF-CNN (*Cai et al., 2021*) | Without | 67.2 | 81.7 | 84.7 | 90.5 | 94.6 |
| Ni-AAD (*Cai et al., 2022*) | Without | 73.1 | 82.8 | 87.1 | 91.2 | – |
| MBSSFCC (*Jiang, Chen & Jin, 2022*) | Without | 80.7 | 89.2 | 91.5 | 93.8 | – |
| AGSLnet (*Zeng, Cai & Xie, 2024*) | Without | 88.1 | 93.6 | 94.1 | – | – |
| SSF+AD-GAN (auditory-GAN) | Without | 79.3 | 84.9 | 89.5 | 96.8 | 98.5 |

**The superiority of the AD-GAN model over traditional data augmentation techniques:** Detection performance with different data augmentation techniques is analyzed, and their effectiveness is compared with the AD-GAN model. The dataset is augmented using traditional data augmentation techniques, specifically rotation and scaling. This data augmentation increases the size of the dataset. The augmented images are directly input to the CNN model without involving the generated images from the GAN model.

The performance achieved with different data augmentations is given in Table 7. In the first experiment, original training data (without any augmentation) are used for model training. In the second experiment, the rotated images (using preset values of 45°, 90°, 135°, 180°, and 220°) are used for the training process. In the third experiment, the scaled images (using nearest-neighbor interpolation (*Rukundo & Cao, 2012*) with a zooming factor of 0.85) are used. In the fourth experiment, all the original and augmented images obtained with different augmentations are used in training. These experiments show that augmentation increases the performance of the model compared to using original data alone. The model trained on the augmented set of images shows better results in comparison to the model trained on original images. However, the detection performance obtained with GAN-generated data (reported in Table 6) is superior compared to traditional augmentation techniques (reported in Table 7).

**Detection performance across different window sizes:** Our auditory-GAN model determines between right and left spatial attention considering each EEG data segment. Since the test is evenly distributed between right-left attention, there is 50% chance-level. The average per-subject accuracy across five decision windows (0.1, 1, 2, 5, 10) is reported in Table 8. Our auditory-GAN model attains an accuracy of 79.3% (SD: 3.77) for 0.1 s, 84.9% (SD: 4.22) for 1 s, 89.5% (SD: 5.12) for 2 s, 96.8% (SD: 4.41) for 5 s and 98.5% (SD: 3.98) for the 10 s decision window. From the results, it's apparent that detection accuracy for longer decision windows is higher than the smaller length of decision windows. To carry out the fair comparison, the results are compared with the previous state-of-the-art (*Cai et al., 2021*; *O'Sullivan et al., 2015*; *Cai et al., 2022*; *Vandecappelle et al., 2021*; *Jiang, Chen & Jin, 2022*; *Zeng, Cai & Xie, 2024*).

The linear decoder (*O'Sullivan et al., 2015*) is considered to be one of the best linear auditory attention detection decoders to date (*Geirnaert, Francart & Bertrand, 2020*). The linear decoder (*O'Sullivan et al., 2015*) is stimulus (speech envelope) reconstruction model that involves approximating the envelope of the attended speech using EEG signals. The linear decoder requires speech stimuli as the reference, and our auditory-GAN decodes human attention purely from the brain signals themselves. Other tested nonlinear auditory attention models also used direct classification without explicitly reconstructing the speech envelope (stimulus).

It is significant to notice that our model shows a more consistent accuracy trend across different window sizes compared to other baseline models *i.e.*, Ni-ADD (*Cai et al., 2022*), CNN (*Vandecappelle et al., 2021*), and SSF-CNN (*Cai et al., 2021*) with a fewer number of exceptions. This accuracy difference becomes more noticeable, particularly with shorter window lengths. Specifically, there is a notable decrease in accuracy for the linear decoder, *i.e.*, 58.1% with 1 s decision window, while the decrease in accuracy of the SSF-CNN model (81.7%) is less compared to the linear model, while our AD-GAN model retains performance level reasonably well (84.9%). The results shown in Table 8 demonstrate that variation in decoding accuracy with different window lengths is consistent with the literature (*Cai et al., 2021*; *O'Sullivan et al., 2015*; *Cai et al., 2022*; *Vandecappelle et al., 2021*; *Jiang, Chen & Jin, 2022*; *Zeng, Cai & Xie, 2024*).

The comparative analysis of the detection performance of the proposed AD-GAN system with benchmark techniques has been given in Table 8. In addition, to confirm that the proposed AD-GAN model significantly improves over its counterpart, we performed a paired t-test, $p = 0.028$. The major difference between the designed AD-GAN model and SSF-CNN lies in the data augmentation strategy, the performance improvement is due to the training of the classifier with the massive amount of artificial training data generated by the deep GAN model. Since the time delay needed by humans to shift their attention during decision-making processes is 1 s, therefore, we aimed to explore the limits concerning decision window lengths. To check the effect of window length on model performance, we tested the AD-GAN and SSF-CNN models with window lengths shorter than 1 s (0.1 ms). It is important to notice that the AD-GAN model not only surpasses the SSF-CNN and NI-AAD on shorter window lengths of 0.1 s but also on the window lengths of 1 and 2 s. Results show that the model presents good accuracy on the KUL dataset with all decision windows (0.1 to 10 s). Since it eliminates the need for additional data for training, the proposed AD-GAN model is considered very viable for neuro-steered hearing aids that can be used in other everyday applications and remains an appropriate solution even for the requirement of low-latency.

**The detection accuracies of the auditory-GAN system on DTU dataset** (*Fuglsang, Wong & Hjortkjær 2018*): To further evaluate the model's AAD ability, performance was tested with the DTU dataset. DTU dataset was acquired from 18 normal-hearing individuals. The two speech streams were presented to the subjects lateralized at −60° and +60° along the azimuth direction. A BioSemi Active system was used to record the 64-channel EEG signals at a sampling rate of 512 Hz. Fifty minutes of data were recorded for each subject, resulting in a total of 15 h of EEG data collected from all subjects. For model

**Table 9 Detection accuracy (%) across five different decision window sizes on the DTU dataset (*Fuglsang, Wong & Hjortkjær 2018*) of 64 channels.**

| Model | Auditory stimulus | Decision window | | | | |
|---|---|---|---|---|---|---|
| | | 0.1 s | 1 s | 2 s | 5 s | 10 s |
| CNN (*Vandecappelle et al., 2021*) | Without | – | 69.2 | 71.2 | 71.9 | – |
| SSF-CNN (*Cai et al., 2021*) | Without | 62.2 | 63.4 | 66.1 | 70.0 | – |
| Ni-AAD (*Cai et al., 2022*) | Without | 59.7 | 61.6 | 63.2 | 61.5 | – |
| MBSSFCC (*Jiang, Chen & Jin, 2022*) | Without | 66.2 | 76.8 | 82.8 | 82.87 | – |
| AGSLnet (*Zeng, Cai & Xie, 2024*) | Without | 76.1 | 80.8 | 81.3 | – | – |
| SSF+AD-GAN (auditory-GAN) | Without | 71.1 | 76.3 | 80.2 | 88.3 | 90.1 |

**Table 10 Attention detection accuracy (%) of the proposed auditory-GAN model using various machine vision classifiers.**

| | Auditory-GAN-RF | Auditory-GAN-SVM | Auditory-GAN-SoftMax | Auditory-GAN-FC |
|---|---|---|---|---|
| Precision | 88.9 | 91.5 | 93.7 | 97.7 |
| Recall | 90.2 | 90.2 | 94.8 | 98.8 |
| F Score | 89.5 | 90.8 | 94.2 | 98.2 |
| Accuracy | 91.4 | 93.5 | 96.4 | 98.5 |

**Table 11 Detection accuracy (%) across five different decision window sizes on the KUL dataset of 32 channel.**

| Model | Auditory stimulus | Decision window | | | | |
|---|---|---|---|---|---|---|
| | | 0.1 s | 1 s | 2 s | 5 s | 10 s |
| SSF-CNN (*Cai et al., 2021*) | Without | – | 76.1 | 80.1 | 86.2 | 89.4 |
| **SSF+AD-GAN (auditory-GAN)** | Without | – | 79.8 | 85.6 | 91.3 | 95.3 |

training on the DTU dataset, the same experimental settings are followed as was set for KUL dataset. Similar to KUL dataset, the accuracies of audiotry-GAN system on the DTU dataset (given in Table 9) is compared with other recently proposed state-of-the-art methods (*Cai et al., 2021*, *2022*; *O'Sullivan et al., 2015*; *Vandecappelle et al., 2021*; *Jiang, Chen & Jin, 2022*; *Zeng, Cai & Xie, 2024*) using different decision windows. In comparison to the KUL dataset, the performance on DTU is slightly low. This analysis shows that the AD-GAN model outperforms the other techniques Specifically, for 5 and 10 s decision windows.

In Tables 8 and 9, for comparative analysis, the SSF-CNN model (*Cai et al., 2021*) has been reimplanted with our experimental setup while the other methods reported with their original setting.

**The detection accuracies of auditory-GAN system with various machine vision classifiers:** To evaluate the impact of different machine vision classifiers on the detection

performance of the proposed model, distinct experiments are performed with four state-of-the-art machine vision classifiers, namely, RF (*Pal, 2005*), support vector machine (SVM) (*Burges, 1998*), Softmax (*Peng et al., 2017*), and fully connected (FC) layers in the discriminator model. The detection results with various classifiers are given in Table 10. The results in Table 10 reveal that the highest accuracy is achieved when FC layers are employed with our model in comparison to other tested classifiers. Therefore, we considered FC layers the most robust classifier to use in the final auditory-GAN model.

**Performance using 32 channels of EEG data:** The results discussed above are based on 64-channel dataset of EEG. A smaller sum of electrodes in EEG signal equalization has many advantages. Therefore, 32-channel EEG data, acquired by reducing electrodes from 64 to 32 channels (*Das, Francart & Bertrand, 2019*), is also used in our experiments. The performance of the auditory-GAN system on 32-channel data is given in Table 11. The auditory-GAN achieves 79.8% (SD: 5.22), 85.6% (SD: 5.26), 91.3% (SD: 5.04), and 92.4% (SD: 6.45) for 1, 2, 5, and 10 s decision windows of 32 channels data, respectively. It is observed that accuracy achieved on 32-channel data is moderately low as compared to 64-channel data. However, the auditory-GAN outperforms the formerly proposed SSF-CNN technique using 32-channel data by a large margin of about 5% on overall decision windows and compares constructively for 64-channel data.

**Model robustness against input noise perturbations:** EEG signals are acquired using various types of equipment and in different laboratories, leading to contamination with various kinds of noise. These noises can occur during the signal acquisition, transmission, and storage processes. Such noise perturbations can lead to misclassification by deep models. To test the robustness of the proposed AD-GAN model against input noise perturbations, including multiplicative and additive noise, we introduced multiple levels of Gaussian noise signals (*Kusk & Lysdahlgaard, 2023*) to the test EEG images. Noise perturbation can be represented as:

$$\overline{X} = n.X + m.randn(x_1, x_2, 3) \tag{7}$$

where $\overline{X}$ represents the corrupted image of original image $X$ bearing size $x_1 \times x_2 \times 3$. The *randn*() denotes the Gaussian noise function. The levels of noise are set as $n = 0.5$ and $m = 1, 3, 5, 7, 9, 11$. Noise robustness test is performed while adding noise at different levels to the input EEG images to contaminate them. In contaminated or corrupted images, the overstepping pixel values are restricted to the bounds [0, 255]. The box plots given in Table 12 show the strength of the developed auditory-GAN against different Gaussian noise levels. In the next step, contaminated EEG test images are fed to the AD-GAN model, and output accuracy is observed. The results shown in Table 13 reveal that there is a trivial decrease in accuracy for various noise levels which shows the strong robustness of AD-GAN against input noise perturbations.

**Performance of AD-GAN network for intra-subject auditory attention detection:** To perform intra-subject auditory attention detection, the training and test data for machine learning came from the same subject. We conducted intra-subject AAD experiments with the AD-GAN network using 16 subjects from the KUL AAD dataset. The AD-GAN network was trained subject by subject. The data were split into five folds, and model

**Table 12** Results of noise robustness test (A) accuracy on 64-channel EEG images contaminated with different noise values, and (B) 32-channel EEG images contaminated with various noise levels.

(a)

(b)

**Table 13 Robustness of proposed auditory-GAN against different noise levels $n = 0.5$ and m = 1, 3, 5, 7, 9, 11. With decision window 10 s.**

| Noise levels | 64-channel images contaminated with various noise levels | 32-channel images contaminated with various noise levels |
|---|---|---|
| | Accuracy (%) | Accuracy (%) |
| Image without noise | 98.5 | 92.4 |
| m = 1 | 97.9 | 90.1 |
| m = 3 | 97.1 | 88.3 |
| m = 5 | 96.8 | 87.2 |
| m = 7 | 95.4 | 86.5 |
| m = 9 | 95.1 | 85.6 |

**Table 14 Performance accuracy (%) of the AD-GAN network for the 16 subjects in the KUL AAD dataset.**

| | S1 | S2 | S3 | S4 | S5 | S6 | S7 | S8 | S9 | S10 | S11 | S12 | S13 | S14 | S15 | S16 |
|---|---|---|---|---|---|---|---|---|---|---|---|---|---|---|---|---|
| Precision | 86.2 | 89.5 | 90.8 | 92.6 | 96.7 | 94.4 | 95.1 | 95.7 | 97.1 | 97.3 | 98.7 | 98.2 | 95.6 | 98.7 | 96.4 | 98.6 |
| Recall | 88.5 | 87.2 | 91.2 | 93.7 | 97.2 | 96.3 | 94.8 | 96.5 | 98.3 | 98.2 | 99.1 | 97.8 | 97.8 | 97.8 | 97.7 | 98.7 |
| F Score | 87.3 | 88.3 | 90.9 | 93.1 | 96.9 | 95.3 | 94.9 | 96.0 | 97.6 | 97.7 | 98.8 | 97.9 | 96.6 | 98.2 | 98.2 | 98.6 |
| Accuracy | 90.4 | 92.4 | 94.3 | 95.5 | 98.3 | 95.5 | 96.2 | 97.4 | 98.6 | 98.1 | 97.2 | 99.0 | 95.5 | 98.4 | 96.5 | 97.1 |

**Table 15 McNemar's statistical (*McNemar, 1947*) test using different datasets.**

| Models | KUL dataset (*Das, Francart & Bertrand, 2019*) p-value |
|---|---|
| Linear decoder (*O'Sullivan et al., 2015*) | 0.01 |
| CNN (*Vandecappelle et al., 2021*) | 0.01 |
| SSF-CNN (*Cai et al., 2021*) | 0.04 |
| Ni-AAD (*Cai et al., 2022*) | 0.02 |
| MBSSFCC (*Jiang, Chen & Jin, 2022*) | 0.01 |
| AGSLnet (*Zeng, Cai & Xie, 2024*) | 0.03 |

evaluation was carried out using inner 5-fold cross-validation (*Wong et al., 2018*) for each subject. Specifically, the AD-GAN network generated a total of 5,280 samples per subject, resulting in 84,480 samples for the 10-s window case. In this analysis, the performance of the subject-dependent validation strategy was evaluated by training the model subject by subject. For each subject, the data were randomly split into five folds according to inner five-fold cross-validation (*Wong et al., 2018*). One fold was used for testing, while the remaining four folds were used for training the model. The training process was repeated five times, and average results were calculated. The reported results represent the average performance across all testing folds. Table 14 shows the accuracies of the AD-GAN network for each subject.

**McNemar's statistical test** (*McNemar, 1947*): To check the robustness of the proposed auditory-Gan system in comparison to the other classifiers, the McNemar's statistical test is performed. This test checked whether the designed technique is significantly better than the other methods in terms of error rates. For this test, the value of significance level is set to 0.05. The McNemar's *p*-values from various tests are reported in Table 15. The reported results demonstrate that the proportion of errors is uniform among all compared methods, and differences in accuracies can be ignored. This robustness test demonstrates that auditory-GAN model outperforms the other methods.

## CONCLUSION

The evolution of deep models relies heavily on large amounts of labeled data. However, the high cost associated with data collection makes attention detection with small-scale datasets a challenging task during model training. To address this, this article presents a system for generating artificial EEG data and detecting auditory spatial attention. The proposed Auditory-GAN system utilizes spatial feature maps of EEG signals to meet the need for extensive training data while performing auditory spatial attention detection. The Auditory-GAN strategy achieves encouraging results and surpasses existing models, even with a small decision window of 0.1 s. It has significant applications in fields such as neuro-steered hearing aids, cochlear implants, and speech recognition systems. The investigations into the impact of data augmentation on auditory detection, proposed in this article, could potentially transform the landscape of auditory attention detection research.

Despite EEG data sparsity challenges, the AD-GAN is a promising solution for the AAD task. The AD-GAN is a promising solution for AAD tasks. The noise robustness analysis also demonstrates the strong generalization of the designed techniques to accept diverse datasets with inevitable noise perturbations.

Despite EEG data sparsity challenges, the AD-GAN is a promising solution for the AAD task. The noise robustness analysis also demonstrates the strong generalization of the designed techniques to handle diverse datasets with inevitable noise perturbations. However, the auditory-GAN system has few limitations. The EEG datasets available online may not encompass the full range of AAD scenarios, which could affect the model's performance in unseen environments. Experimental analysis shows that model performance varies across different datasets; for example, the detection accuracy for the KUL dataset is higher compared to the DTU dataset. This variation is due to differences in recording equipment, EEG signal characteristics, environmental noise, and subject variability. Detection accuracy is also sensitive to the choice of hyperparameters.

Further research on designing robust generative models with efficient training strategies can improve auditory attention detection. In the future, the validity of the model can be tested in practical applications by incorporating real-world data. Moreover, the robustness of the proposed model can be further assessed across various datasets by using additional data preprocessing techniques. Considering the effectiveness of deep learning techniques in EEG analysis, researchers might also explore data augmentation with other deep models, such as GCNs.

### Funding

This work was supported by the National Natural Science Foundation of China under Grant 62176102; the Joint Fund of Basic and Applied Basic Research Fund of Guangdong Province under Grant No. 2020A1515110498 and Grant No. 2020A1515140109; the Fund aimed at Improving Scientific Research Capability of Key Construction Disciplines in Guangdong Province, specifically for the project "Light-Weight Federal Learning Paradigm and its Application," under Grant 2022ZDJS058; and the Professorial and Doctoral Scientific Research Foundation of Huizhou University under Grant 2020JB058. There was no additional external funding received for this study. The funders had a role in study design, data collection and analysis, decision to publish, or preparation of the manuscript.

### Grant Disclosures

The following grant information was disclosed by the authors:
National Natural Science Foundation of China: 62176102.
Joint Fund of Basic and Applied Basic Research Fund of Guangdong Province:
2020A1515110498, 2020A1515140109.
Light-Weight Federal Learning Paradigm and its Application: 2022ZDJS058.
Professorial and Doctoral Scientific Research Foundation of Huizhou University:
2020JB058.

### Competing Interests

The authors declare that they have no competing interests.

### Author Contributions

- Tasleem Kausar conceived and designed the experiments, prepared figures and/or tables, and approved the final draft.
- Yun Lu performed the experiments, prepared figures and/or tables, and approved the final draft.
- Muhammad Awais Asghar analyzed the data, prepared figures and/or tables, and approved the final draft.
- Adeeba Kausar performed the computation work, authored or reviewed drafts of the article, and approved the final draft.
- Siqi Cai analyzed the data, authored or reviewed drafts of the article, and approved the final draft.
- Saeed Ahmed performed the computation work, authored or reviewed drafts of the article, and approved the final draft.
- Ahmad Almogren analyzed the data, authored or reviewed drafts of the article, and approved the final draft.

### Data Availability

The code is available at Zenodo: tasleem-hello. (2024). tasleem-hello/Auditory-GAN-: AD-Gan (1.0). Zenodo. https://doi.org/10.5281/zenodo.13334755.

The data is available at Zenodo: Das, N., Francart, T., & Bertrand, A. (2019). Auditory Attention Detection Dataset KULeuven (2.0) [Data set]. Zenodo. https://doi.org/10.5281/zenodo.4004271.

## Supplemental Information

Supplemental information for this article can be found online at http://dx.doi.org/10.7717/peerj-cs.2394#supplemental-information.

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
