# Peer review of "Auditory-GAN: deep learning framework for improved auditory spatial attention detection"

_PeerJ Computer Science, doi:10.7717/peerj-cs.2394_

## Round 0.1 · original submission · Major Revisions

Dear authors,

Thank you for submitting your article. Reviewers have now commented on your article and suggest major revisions. We do encourage you to address the concerns and criticisms of the reviewers and resubmit your article once you have updated it accordingly. When submitting the revised version of your article, it will be better to address the following:

1. Equations should be used with correct equation number. Please do not use “as follows”, “given as”, etc. Explanation of the equations should also be checked. All variables should be written in italic as in the equations. Their definitions and boundaries should be defined. Necessary references should be given.
2. Pros and cons of the method should be clarified. What are the limitation(s) methodology(ies) adopted in this work? Please indicate practical advantages, and discuss research limitations.
3. Please include future research directions.

Best wishes,

Reviewer 1 ·

Basic reporting

This manuscript ID-100889 introduces Auditory-GAN, a novel deep learning framework aimed at enhancing auditory spatial attention detection from EEG signals. It addresses challenges posed by limited EEG data and the need for low-latency detection. The framework consists of an SSF module for extracting spatial feature maps and an AD-GAN classifier for data augmentation and attention detection. Evaluated on the KUL dataset, Auditory-GAN demonstrated high accuracy, achieving a 98.5% detection rate for a 10-second decision window. The model's performance was superior to existing state-of-the-art methods, especially in handling EEG data with varying channel numbers. It was a pleasure reviewing this work and I can recommend it for publication in PeerJ Computer Science after a major revision. I respectfully refer the authors to my comments below.

Experimental design

1. The English needs to be revised throughout. The authors should pay attention to the spelling and grammar throughout this work. I would only respectfully recommend that the authors perform this revision or seek the help of someone who can aid the authors.
2. (References) Please adjust the style of all the references to meet the PeerJ Computer Science requirement.
3. (Page 19) The original figure 1 is not clear. Please redraw this figure clearly. Add some word in this figure, and indicate the stage and modules.
4. The introduction could be strengthened by clarifying how Auditory-GAN advances existing stimulus reconstruction-based methods. A comparative table highlighting their limitations would be useful.
5. While the KUL dataset is a good start, validating the framework on additional EEG datasets like BrainInvaders would enhance generalizability.
6. (In Section 1) The original sentence is suggested to revise as “The advent of deep learning models, particularly convolutional neural networks (CNNs) [*], …”. ([1] "edmf: efficient deep matrix factorization with review feature learning for industrial recommender system," ieee tii, 2022; [2] "multi-perspective social recommendation method with graph representation learning," neurocomputing, 2022. [3] "carm: confidence-aware recommender model via review representation learning and historical rating behavior in the online platforms," neurocomputing, 2021.)

Validity of the findings

7. (Section 1 Introduction) The reviewer hopes the introduction section in this paper can introduce more studies in recent years. The reviewer suggests authors don't list a lot of related tasks directly. It is better to select some representative and related literature or models to introduce with certain logic. For example, the latter model is an improvement on one aspect of the former model.
8. A deeper analysis of the impact of hyperparameters, such as learning rate and batch size, on the AD-GAN's performance would provide valuable insights.
9. Comparing Auditory-GAN with more recent deep learning-based auditory attention detection methods would make the evaluation more comprehensive.
10. (Figure 3-4) Experimental pictures or tables should be described and the results should be analyzed in the picture description so that readers can clearly know the meaning without looking at the body.

Additional comments

11. (In Section 2.2) The original sentence is suggested to revise as “Data augmentation is widely used in machine learning and plays a vital role in improving performance [*], reducing overfitting, and reinforcing the generalization capabilities, …” ("orientation cues-aware facial relationship representation for head pose estimation via transformer," ieee tip, 2023.; "transifc: invariant cues-aware feature concentration learning for efficient fine-grained bird image classification," ieee tmm, 2023.; "tokenhpe: learning orientation tokens for efficient head pose estimation via transformers," in 2023 ieee/cvf cvpr 2023)
12. (Table 3-4) All the values in this table should be with same data accuracy. The number of data after the decimal point are the same. Please check other Tables and section.
13. The authors are suggested to add some Auditory Spatial Attention Detection experiments with the methods proposed in other literatures, then compare these results with yours, rather than just comparing the methods proposed by yourself on different models.
14. Discuss the pros and cons of the proposed Auditory Spatial Attention Detection models.

My overall impression of this manuscript is that it is in general well-organized. The work seems interesting and the technical contributions are solid. I would like to check the revised manuscript again.

Reviewer 2 ·

Basic reporting

The authors proposed a complete deep auditory generative adversarial network auxiliary, i.e., auditory-GAN, to handle the challenges while generating EEG data and
executing auditory spatial detection. The proposed auditory-GAN system is composed of a spectro-spatial feature extraction module and an auditory generative adversarial
network auxiliary classiûer. The SSF module extracts the spatial feature maps via learning the topographic speciûcity of alpha power from EEG signals. The designed AD-GAN network addresses the need for extensive training data by synthesizing augmented versions of original EEG data.

Experimental design

It is well designed.

Validity of the findings

The proposed method is verified under the KUL dataset. It is shown that the proposed auditory-GAN can produce convincing EEG data and achieve 98.5% spatial attention.

Additional comments

It is interesting and original, well-organized and can be accepted after minor revision.
There are no necessary discussion and explanation on the simulation results.

Can quantum computing helpful to tackle this problem? Please discuss it.

Where are the limitations of your study? The limitations of a study allows the readers to understand better under which conditions the results should be interpreted.
Future work should be discussed.
The conclusion should be more informative.

·

Basic reporting

The manuscript presents a novel deep learning framework, named Auditory-GAN, designed to improve auditory spatial attention detection from EEG signals. This approach combines a spectro-spatial feature extraction (SSF) module with an auditory generative adversarial network auxiliary (AD-GAN) classifier. The proposed system aims to address challenges related to data scarcity and low-latency detection of auditory attention.
The manuscript is generally well-written and conveys the research clearly. Technical terms are used appropriately, and the structure of the sentences is mostly professional. The manuscript provides a thorough review of the relevant literature, including recent advances and challenges in auditory attention detection using EEG. References are well-chosen and give a comprehensive background to the field.
The manuscript is well-organized, with a clear structure that includes sections on methodology, results, and discussion. Figures and tables are effectively used to illustrate the results and support the text. The codes is well written although it would be better if it is more organized and splitted into modules.

I would suggest the authors to expand the discussion to include a broader range of data augmentation techniques and compare their effectiveness with the proposed GAN-based method. It is also suggested to include a detailed description of the hyperparameter tuning process and discuss potential biases in the data augmentation process.

Experimental design

Strengths: The experimental design of the manuscript is well-structured and comprehensive. The use of a novel deep learning framework, Auditory-GAN, for generating augmented EEG data to improve auditory spatial attention detection is innovative and aligns well with the study’s objectives. The inclusion of various decision windows and noise robustness tests strengthens the validity of the experimental results. While the design is robust, the manuscript would benefit from a more detailed explanation of the choice of specific experimental parameters, such as the decision to use certain window lengths and noise levels. This would provide a clearer rationale for the experimental setup.
The research presented is original and fits well within the aims and scope of a journal focused on advancements in deep learning and its applications in biomedical signal processing. The study addresses a significant challenge in the field—data scarcity in EEG-based auditory attention detection—and provides a novel solution through the use of GANs.
The research question is well-defined, relevant, and meaningful. The manuscript clearly states the challenges in auditory attention detection and proposes the Auditory-GAN framework as a solution. The objectives are clearly outlined, and the significance of the research is well-justified.

Validity of the findings

The findings presented in the manuscript are valid and robust. The use of multiple metrics to evaluate the performance of the Auditory-GAN framework ensures that the results are statistically sound and well-controlled. The experimental results are convincing, showing significant improvements over state-of-the-art methods in terms of accuracy and robustness. The conclusions are well-stated and linked to the original research question. The manuscript provides a detailed methodology, and the availability of the model code on GitHub encourages replication. The rationale for the study is clearly stated, and the benefits to the literature, particularly in addressing the data scarcity problem, are evident. The data used for validation is robust, and the statistical analysis is thorough. The data used for validation is robust. But it is recommended to add more statistical analysis to validate the findings.

---

## Round 0.2 · accepted · Accept

Dear authors,

Thank you for clearly addressing all the reviewers' comments. I confirm that the quality of your paper has been improved. The paper now appears to be ready for publication in light of this revision.

Best wishes,

Reviewer 1 ·

Basic reporting

The revised manuscript is improved compared to the former version. My previous comments are well addressed, and the presentation is improved significantly. The composition pattern and some other ideas are well elaborated, making them clearer. Overall, I tend to accept this manuscript.

Experimental design

The revised manuscript is improved compared to the former version. My previous comments are well addressed, and the presentation is improved significantly. The composition pattern and some other ideas are well elaborated, making them clearer. Overall, I tend to accept this manuscript.

Validity of the findings

The revised manuscript is improved compared to the former version. My previous comments are well addressed, and the presentation is improved significantly. The composition pattern and some other ideas are well elaborated, making them clearer. Overall, I tend to accept this manuscript.

Additional comments

The revised manuscript is improved compared to the former version. My previous comments are well addressed, and the presentation is improved significantly. The composition pattern and some other ideas are well elaborated, making them clearer. Overall, I tend to accept this manuscript.

Reviewer 2 ·

Basic reporting

The manuscript is revised well and all the comments are fully considered.

Experimental design

It is well designed and the results are explained.

Validity of the findings

It sounds well.

Additional comments

None.

·

Basic reporting

The manuscript is generally written in clear, professional way. The manuscript is well-referenced and situates the study within the context of current research. However, expanding the discussion on recent advancements in GAN-based EEG signal generation would strengthen the argument. Including comparisons with more contemporary methods, especially those beyond the auditory-GAN framework, would provide a more robust context. The figures are of good quality, and the raw data provided is appropriate for verification.

Experimental design

The research question is well defined, addressing a clear gap in auditory attention detection from EEG signals. The scope fits within the journal’s aims, and the integration of generative adversarial networks (GANs) for data augmentation is innovative. However, the novelty of the GAN approach should be highlighted more prominently, especially in comparison to previous state-of-the-art methods.
The methods are described in good detail, allowing for replication. However, some improvements can be made in structuring the methodology. For instance, the data preparation section could benefit from a clearer breakdown of steps, including preprocessing, data augmentation, and evaluation. The inclusion of more detailed pseudocode or flowcharts for the AD-GAN architecture might improve clarity for other researchers trying to replicate the work.
Data and Replication: Sufficient detail is provided for replication.

Validity of the findings

The manuscript offers a thorough statistical evaluation of the results, which are both robust and significant. The comparison with conventional GAN methods (Tables 4 and 7) demonstrates the clear advantage of the auditory-GAN system. However, a more in-depth discussion of the limitations and potential pitfalls of this approach is needed, particularly in terms of generalizability across datasets. The issue of dataset noise and variability could be expanded upon to provide a more balanced view.
The conclusions drawn from the findings are well-supported by the data. However, the manuscript could benefit from a more explicit discussion of future research directions and real-world applications, particularly how this method could be generalized to other EEG-based detection tasks or integrated into practical systems such as hearing aids or neurofeedback devices.